# Do Laboratory Blood Tests Change Medical Care in Patients Hospitalized with Community-Acquired Pneumonia?

**DOI:** 10.3390/diagnostics14030302

**Published:** 2024-01-30

**Authors:** Zvi Shimoni, Muhamad Gazi, Paul Froom

**Affiliations:** 1The Adelson School of Medicine, Ariel University, Ariel 4077625, Israel; zshimoni@laniado.org.il; 2Sanz Medical Center, Laniado Hospital, Netanya 4244916, Israel; mohamd.gazi82@gmail.com; 3Clinical Utility Department, Sanz Medical Center, Laniado Hospital, Netanya 4244916, Israel; 4School of Public Health, University of Tel Aviv, Tel Aviv 6997801, Israel

**Keywords:** over-testing, internal medicine, medical care, community-acquired pneumonia

## Abstract

*Background and Objectives*: The prevalence of inappropriate laboratory testing is believed to be high, but only a limited number of studies have reviewed medical charts to determine whether tests impact medical care. *Materials and Methods*: From the electronic database, we selected 500 consecutive patients with community-acquired pneumonia who were hospitalized between January 2020 and October 2021. We excluded eight patients who had COVID-19, but were not identified in the database, and were only identified after chart review. To assess the impact of tests on medical care, we conducted a thorough review of the patients’ charts. *Results*: The age of the patients was 78 ± 16 years, with 42.3% female (n = 208) hospitalized for a median of 4 days (25–75%, 3–6 days). There were 27957 laboratory test results during 2690 hospital days (10.4 tests per day of hospitalization). Of the 2997 tests carried out on admission 5.7% (n = 170) resulted in changes of medical care in 34.5% (170/492) of the patients, nearly all from the results of electrolytes, renal function tests, and serum glucose measurements. Tests that did not lead to any decision on medical care included 75.8% (7181/9478) on admission and 86.0% (15,898/18,479) on repetitive testing, i.e., repetitive testing accounted for 68.9% (15,898/23,079) of tests that did not change medical care. By excluding tests that did not change medical care, the overall testing rate would decrease by 82.6% (23,079/27,947), and from 10.4 tests per day to 2.1 tests per day. *Conclusions*: We conclude that the estimate of the overuse of laboratory testing, which includes all testing that does not change patient care, is much higher than reported using other methodologies. Most of the overuse was from repetitive testing that included unnecessary testing in patients without admission test results that changed medical care. Further investigation is needed to determine if these findings can be applied to patients with diverse health conditions and in different healthcare settings.

## 1. Introduction

A commitment to thoughtful laboratory test ordering could help support a culture of high-value care and stop the needless day-after-day sticking of hospitalized patients [1]. Attempts to reduce unnecessary testing include the Choosing Wisely campaign that began in the United States in 2012 and has spread to more than 20 countries worldwide [2]. The definitions of laboratory over-testing, however, have been varied, and to effectively address unnecessary testing, it is crucial to clearly define what constitutes inappropriate utilization [3,4].

High rates of inappropriate testing have been reported using tools that do not require chart review, including the mean abnormal rate, measurement of redundant testing, and assessment of inter-clinician variability in testing patterns [5]. One study measured the rate of tests of doubtful clinical importance by analyzing the failure to reorder tests that were sent but not reported [6]. This study estimated that approximately 60–70% of test results ordered in a high-throughput laboratory may be potentially inappropriate or of doubtful clinical importance. There are two reports of inappropriate testing in medical hospitalizations identified by chart review. In an academic medical department in Australia [7], 28.6% of tests on admission and 69.3% of tests on follow-up provided no meaningful contribution to clinical management defined as not altering medical care or being relevant to the patient’s symptoms, or potentially dictating the ordering new laboratory tests. In another study of 177 patients hospitalized at two Maryland hospitals, 49% of admission and follow-up laboratory or imaging tests did not change medical management [8]. However, rates of over-testing may be higher if defined strictly as laboratory tests that do not change medical therapy.

In the following historical prospective study, we reviewed the medical records of patients hospitalized with community-acquired pneumonia to determine the proportion of laboratory blood tests that changed medical therapy.

## 2. Materials and Methods

The study population included all acutely admitted adult patients in 2020 and January–October 2021 to the three internal medicine departments at the Laniado Hospital, a regional hospital with 400 beds in Israel. Patients admitted electively or to intensive care units were not included.

In this historical prospective study, we selected 500 consecutive patients from the electronic database with community-acquired pneumonia diagnosed by the admission chest X-ray, representing 5.1% of hospitalizations to internal medicine departments. The chest X-ray was conducted because of fever with or without accompanying respiratory tract symptoms. We also excluded patients admitted on a respirator, with shock or multiorgan dysfunction, and those hospitalized with COVID-19 or with aspiration pneumonia. The hospital medical director is responsible for hospital policy, reviews laboratory testing, and meets periodically with physicians and nurses during the daily morning reviews of patient charts. Decisions on laboratory testing are made during the meetings, during rounds, or independently by physicians. Except for troponin testing, there is no hospital policy on recommendations for laboratory blood testing. The hospital is not reimbursed for laboratory tests, but according to days of hospitalization. All patients are insured and do not pay for the costs of hospitalization. Results are accessed online and physicians contacted by phone if there is a critical test result. Data downloaded from the electronic database included age, sex, days of hospitalization and the number of high throughput admission test results (Table 1 and Table 2). Less common laboratory tests were recorded during chart review. The electronic patient records include chief complaints on admission to the emergency department, admission and discharge summaries, medical treatments, and daily physician and nurse shift reports. Tests are commonly coupled (electrolytes, renal function tests, uric acid and calcium, liver function tests, and lipid tests).

The impact of tests on medical management was assessed by chart review and categorized as those that influenced or did not influence decisions about medical care, including intravenous fluids, hypoglycemic drug treatment, or antibiotic therapy. Repeat tests performed in those with admission tests that resulted in changes in intravenous therapy or hypoglycemic dose, or with admission hemoglobin concentrations < 8 gm/dL, were considered to influence decisions on medical care.

### Statistical Analysis

We determined the total number of tests per day of hospitalization, and the proportion of admission, follow-up, and total tests that influenced medical therapy.

## 3. Results

Eight patients with COVID-19 who were not identified in the database, but only after chart review, were excluded. The most common chest radiographic abnormality was interstitial infiltrates, 27.6% (136/492) had lobar pneumonia, which was bilateral or multi-lobar in 106. The mean age of the patients was 78 ± 16 years, 42.3% were female (N = 208), and they were hospitalized for a median of 4 days (25–75%, 3–6 days).

There were 27,957 test results during 2690 hospital days (10.4 tests per day of hospitalization). Seven tests that influenced decisions on medical care included electrolytes, renal function tests, hemoglobin, serum glucose, and Q fever antibodies (Table 1). Electrolytes, renal function tests, and serum glucose were often outside the reference range and led to intravenous fluid and hypoglycemic agent adjustments in 93 patients and 71 patients respectively (Table 1). Three patients received blood transfusions due to transfusion-dependent myelodysplastic syndrome, active gastrointestinal bleeding, and chronic bleeding due to anticoagulant therapy. Two patients had an earlier-than-scheduled hemodialysis, and one patient with Q fever antibodies had a change in antibiotic therapy. Of the 2997 tests performed on admission, 5.7% (n = 170) led to changes in medical care in 34.5% (170/492) of the patients.

Of the total tests, 51.7% (14,480/27,957) did not change medical management either on admission or on follow-up (Table 2), including 7299 follow-up tests (50.4%) that led to 28 negative viral antibody tests, and one elevated troponin test on admission and follow-up without changes in the electrocardiogram or chest pain leading to a percutaneous coronary arteriography without the placement of a stent. Blood cultures were positive in 27 patients, but did not change antibiotic therapy.

Of the tests that changed medical decisions on admission, there were 11,180 repeat tests, of which only 23.0% (n = 2581) influenced medical therapy (Table 3). There were two patients with changes in therapy due to repeat test results. Both patients had a drop in hemoglobin tested because of acute gastrointestinal bleeding during hospitalization.

In summary, tests that did not lead to any decision on medical care included 75.8% (7181/9478) on admission and 86.0% (15,898/18,479) on repetitive testing (Table 4). Therefore, including only orders required to identify all changes in medical therapy would decrease testing by 82.6% (23,079/27,947) and the rate of orders from 10.4 to 2.1 tests per day. Repeat tests accounted for 68.9% 15,898/23,079 of tests that did not change medical care.

The number of complete blood counts (CBC) and electrolytes per day of hospitalization was 0.71 and 1.01, respectively, which decreased to 0.26 and 0.38 for tests that changed medical therapy.

## 4. Discussion

The primary outcome of this investigation shows that a significant proportion, 82.6%, of laboratory tests had no impact on the medical management of patients diagnosed with community-acquired pneumonia. This percentage is notably higher than the rates of over-testing observed in previous studies of internal medicine patients [7,8] and in a meta-analysis encompassing various tests, cohorts, and methodologies where the estimated rate of over-testing was around 44% [9]. In one third of the patients, electrolyte levels, renal function tests, serum glucose, hemoglobin, and antibodies to Q fever influenced medical therapy. However, retesting did not change medical care in patients whose admission tests did not alter treatment. Consequently, these findings support the recommendation against biochemical retesting in clinically stable patients [10].

The strength of this study is that a change in medical care defined appropriate laboratory testing, not dependent on preconceived notions. Our findings align with expert opinions on testing for patients with community-acquired pneumonia [11], which do not recommend tests we found to have no impact on medical therapy. These tests include liver functions, prothrombin times, serum lipids, lactate dehydrogenase, calcium, uric acid, creatinine phosphokinase, C-reactive protein, and troponin. Their recommendation for blood cultures is limited to patients with severe disease or treatment for a suspected infection with *Methicillin-resistant Staphylococcus aureus* or *Pseudomonas aeruginosa*. We did not observe changes in antibiotic therapy after a positive blood culture.

Our study has limitations. It is unclear whether the testing rates found in our study are like other geographical areas. However, our rates are consistent with those reported by others in the United States and Europe. For example, in Texas hospitals, patients admitted with bacterial pneumonia had 13 individual units billed for laboratory revenue codes per day of hospitalization [12]. In medical admissions to a Maryland medical center, patients had a mean of 1.5 complete blood count tests per day, even after excluding patients with gastrointestinal bleeding, acute renal insufficiency, congestive heart failure and those transferred to intensive care [13]. In the Netherlands, patients hospitalized in medical wards had 15 individual clinical chemistry tests per day [3].

Secondly, rates of laboratory tests might vary according to presentation. Valencia et al. [12] reported a higher rate in patients with bacterial pneumonia than those with cellulitis. Furthermore, in patients where the admission diagnosis is unclear, other laboratory blood tests can aid in determining the cause of the patient’s presentation. Thirdly, despite resulting in changes in therapy, we cannot assume that all the patients benefited from the changes. There might have been patients who did not require intravenous fluids or a change in their hypoglycemic agent dose. Fourthly, we conservatively considered all follow-up tests in those with changes in treatment on admission as continuing to influence medical therapy.

Lastly, we demonstrated that electrolytes, complete blood counts, renal function tests and serum glucose measurements are essential in some patients acutely admitted to internal medicine departments, but there are other admission tests that will change medical therapy in patients admitted with other presentations; for example, a prothrombin test in a patient admitted with bleeding receiving oral anticoagulation therapy. Thus, a careful medical examination of the patient in the emergency department before ordering tests to determine what tests will change medical therapy is essential. If such an assessment cannot be done before ordering laboratory tests, then admission laboratory tests will need to be routine and not selective.

It is unclear if reducing laboratory testing is safe. A recent metanalysis [14] included studies that assessed interventions including computerized provider order entry, clinical decision support systems/tools, education, auditing with feedback and combinations of those practices. The interventions were variably effective, and apparently not associated with adverse consequences because the rates of the patient-important outcomes assessed (e.g., morbidity, mortality, and length of stay) were not significantly increased. However, demonstrating that the tests did not change medical care is a more sensitive way to measure the absence of adverse consequences. Our findings suggest that if physicians consider whether the test will change patient care before ordering a test, a reduction in laboratory testing by >80% is safe.

It is also unclear how to reduce over-testing due to inexperience or lack of knowledge about the appropriate use of tests [15,16], failure to check previous results, test ordering routines, or fear of errors of omission and litigation. Hoffman et al. claimed that the main driver of over-diagnosis and over-treatment is zero tolerance for error and uncertainty [17]. Other causes include ordering the wrong test, not considering the clinical disutility of over-utilization, and perverse financial incentives. Reimbursement in our hospital is for days hospitalized and not tests done, but financial incentives may influence laboratory testing rates in other settings.

The most important incentive for physicians and nurses to limit laboratory testing is evidence that decreasing over-testing improves the quality and safety of care [18]. It is therefore important to emphasize that excessive phlebotomy can lead to hospital-acquired anemia, unnecessary downstream testing and procedures, and patient discomfort [10,19] whereas commonly laboratory tests do not change medical care. However, the impact of education by itself has limitations [18]. In one study, internal medicine providers were educated through flyers displayed in providers’ offices and periodic email communications reminding them to order daily blood tests only if the results would change patient care (13). The number of CBC tests per day decreased slightly from 1.46 to 1.3. Previously, our research revealed that most troponin test results in the hospital were in patients without chest pain or ischemic electrocardiographic, had no clinical utility, and resulted in inappropriate downstream testing [20]. Based on these findings, we recommended restricting troponin testing to patients with either chest pain or ischemic electrocardiographic changes, resulting in a decrease from 51.5% to 34.6% [21], which closely aligns with the proportion of patients with community-acquired pneumonia who had troponin tests in this study. Nevertheless, the educational intervention achieved partial effectiveness, as only 9.4% of patients tested exhibited chest pain or ischemic electrocardiographic changes.

Hence, it is necessary to implement additional measures to prevent unnecessary invasive interventions observed in one patient during this study. Reviews [14,22,23,24] have not found an intervention that is consistently the most effective, but combined practices are recommended as the best practice to support appropriate laboratory test utilization [14]. This includes audit-and-feedback mechanisms that provide clinicians with data on personal ordering patterns, peer comparisons and targets [19]. The advantage of audit and feedback interventions is that they are widely applicable and do not require modification of the electronic data system. Peer comparisons are particularly effective [19], and nearly all feedback studies have reported some effect [25]. Interventional studies have commonly used two indices [19] provided by the electronic data system to determine utility of the intervention, and to provide feedback to the physicians. Complete blood count and basic metabolic profile test orders are divided by days of hospitalization.

Audit and feedback studies have a varied educational component, affecting the ability to reduce testing. One study gave a lecture and cards emphasizing that retesting is commonly inappropriate [26]. Weekly emails provided peer comparisons of ordering rates and a target value that was 20% lower than before the intervention. CBC rates decreased from 0.90 to 0.52 for the hospital staff and from 0.51 to 0.31 for hospitalists. Others attempted to change testing from routine to thoughtful [27]. Monthly meetings compared physicians’ costs of testing who shared in the savings. During the intervention period, the number of CBC tests per day decreased from 0.84 to 0.73 [27]. In a quality improvement project to reduce overutilization of blood tests by medical residents in a Boston University hospital, CBC testing was discouraged if the hemoglobin, white blood cell count, and platelet count were stable within 24 h of the previous test and if there was no suspicion for bleeding or infection [28]. They held biweekly meetings that compared the rates of CBC and other laboratory tests. The rate of CBC tests per hospital day decreased from 1.56 to 1.45. Another study sent emails reminding physicians to stop ordering tests daily unless indicated and provided monthly reminders with comparative daily rates [29]. The CBC index decreased from 1.06 to 0.91.

Modified electronic medical record systems [14] that require system changes can also improve physician ordering practices by using pop-ups that need to be overridden for repetitive daily laboratory tests and for prior stable test results. Lippi et al. reported that for a limited number of tests alerts are accepted by physicians after an educational intervention [30]. This, however, requires changes in the electronic medical records system that is not possible in all settings.

In our hospital, physicians order laboratory tests routinely and not selectively. We have instituted an educational process focusing on limiting testing that changes medical care. This includes decoupling investigations that will increase time to complete orders but will make orders more specific [18]. We are utilizing our electronic medical records to track the number of specific laboratory tests performed per day of hospitalization, and have initiated the practice of sending weekly departmental comparisons with a target rate based on the results of this study.

## 5. Conclusions

Our research indicates that the assessment of excessive use of laboratory testing, which encompasses all tests that do not impact patient care, is significantly higher than previously estimated. In our setting, the combination of education, auditing with feedback, de-coupling tests, and a modified electronic medical records system if effective could safely reduce laboratory tests by >80% in patients hospitalized with community-acquired pneumonia. A target value for total blood laboratory testing could be as low as 2.1 tests per day, which is useful information to provide for an audit and feedback intervention. Our findings demonstrate that electrolytes, complete blood counts, renal function tests and serum glucose measurements are essential in patients acutely admitted to internal medicine departments. There might be patients who do not require such testing but exclusions of those tests and other tests routinely ordered in the emergency department require a medical evaluation. Studies are needed to determine the rates of over-testing in hospitalized patients with other medical conditions and in different healthcare settings.

## Figures and Tables

**Table 1 diagnostics-14-00302-t001:** Admission tests in 492 patients admitted with pneumonia that changed medical therapy.

Appropriate Tests	Number of Tests	Normal Range	Abnormal Tests	Treatment Change
**Potassium**	492	3.5–5.2	66 (13.2)	19 *
**Sodium**	492	135–145	180 (36.6)	65 *
**Blood urea nitrogen**	492	≤20	270 (54.9)	56 *
**Creatinine**	492	≤1.2	152 (30.9)	56 *
**Total intravenous ***	1968			93 **
**Hemodialysis**				2
**Glucose**	492	≤125	259 (52.6)	71 ***
**Hemoglobin**	492	≥8 gm/dL	45 (9.1)	3 ****
**Q fever antibodies**	45	Negative	1 (2.0)	1 *****
**Total tests**	2997		973 (32.5)	170 (5.7)

* Changes in intravenous fluid administration. ** Multiple abnormal tests were included only once. *** Changes in hypoglycemic agent doses with glucose values of 200 mg/dL or more. **** Blood transfusion. ***** Changed antibiotic treatment.

**Table 2 diagnostics-14-00302-t002:** Tests that did not positively change medical therapy.

Test	Admission	Follow-Up Tests	Disutility
**Alkaline phosphatase**	492	950	
**Albumin**	492	950	
**SGPT ***	492	950	28 negative viral antibodies
**SGOT ***	492	950	
**Prothrombin time**	492	47	
**Cholesterol**	492	24	
**Triglycerides**	492	24	
**Lactate dehydrogenase**	492	874	
**Uric acid**	492	431	
**Hemoglobin A1c**	47	0	
**C-reactive protein**	492	1176	
**Calcium**	492	431	
**Creatinine phosphokinase**	492	463	
**Troponin**	165	1	1 *
**Thyroid stimulating hormone**	64	0	
**Blood cultures**	1001		
**Viral tests ****	0	28	
**Total**	7181	7299	14,480

* One elevated troponin test on admission and on follow-up without changes in the electrocardiogram, and symptoms consistent with pneumonia resulted in a percutaneous coronary arteriography without the placement of a stent. ** Hepatitis C (n = 12), Hepatitis B (N = 12), Epstein–Barr virus (N = 2), Cytomegalovirus (n = 2).

**Table 3 diagnostics-14-00302-t003:** Repeat testing in those with and without changes in medical care.

Tests	Total Repeat Tests	Changes In Medical Care
**Potassium**	2214	523
**Sodium**	2214	523
**Blood urea nitrogen**	1616	419
**Creatinine**	1616	419
**Glucose**	2106	487
**Hemoglobin ***	1414	210
**Total**	11,180	2581(23.0%)

* Two patients had blood transfusions due to an acute gastrointestinal bleed.

**Table 4 diagnostics-14-00302-t004:** Blood tests that influenced medical care.

Tests	Tests N(%)	Follow-Up Tests	Total
**Admission tests that led to decisions on medical care**	2297	11,180	13,477
**No changes in medical care**	2127	8599	
**Changes in medical care**	170	2581	2751
**Admission tests that did not change medical care**	7181	7299	14,480
**Total tests**	9478	18,479	27,957
**Tests required to identify all changes in medical care**	2297 (24.2)	2581 (14.0)	4878 (17.4)

## Data Availability

The data presented in this study are available on request from the corresponding author.

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
