# Peer review of "Do Laboratory Blood Tests Change Medical Care in Patients Hospitalized with Community-Acquired Pneumonia?"

_diagnostics, 2024, doi:10.3390/diagnostics14030302_

Round 1

Reviewer 1 Report

Comments and Suggestions for Authors

Do laboratory blood tests change medical care in patients hos-2 pitalized with community-acquired pneumonia?

An interesting article that discusses the utility or futility of certain analyses in the management of patients with community-acquired pneumonia. Most of the time, additional investigations are conducted, either for a more detailed differential diagnosis or in the case of patients with comorbidities and advanced age.

I have a few observations:

Please correct Ebstein Bar Virus – Epstein Barr – R 91

For the etiological diagnosis of pneumonia, have only Q fever antibodies been performed? What about other analyses such as Mycoplasma antibody, Legionella urinary antigen, and Streptococcus pneumoniae?  - R 110

Has an Multiplex RT PCR been performed from sputum?

Blood cultures were positive in 27 patients but did not  change antibiotic therapy, but the results from sputum cultures?   R 124

"Perhaps it should be separately specified which biochemical analyses did not influence medical care and which bacteriological ones did or did not."

Comments on the Quality of English Language

Minor editing of English language required

Author Response

  1. Please correct Ebstein Bar Virus – Epstein Barr – R 91

1a. Corrected. Thanks, new line 96,

2. For the etiological diagnosis of pneumonia,have only Q fever antibodies been performed for the etiological diagnosis of pneumonia? What about other analyses such as Mycoplasma antibody, Legionella urinary antigen, and Streptococcus pneumoniae?  - R 110

2a. The study only included laboratory blood specimens and didn't include tests of sputum or urine. Streptococcal pneumonia is tested only by blood and sputum cultures, mycoplasma by BioFire system of sputum as well as for other targets, the Legionella antigen is tested in the urine, and influenza in a respiratory specimen. This study did not study changes made in antibiotic therapy based on tests of sputum or urine. Our results in that respect do not add anything to the literature.  

Not to be included in the results; There was a change of antibiotics in 20 patients from 80 positive sputum cultures of 196 cultures sent (3 were de-escalations). Antibiotics were stopped after a PCR was positive for influenza in 3 patients. In one patient there was a de-escalation of the antibiotics after a Biofire test (N=18)  result and none of the 31 patients had a p[positive Legionella urinary antigen test. More appropriate antibiotic therapy was given when a serological test revealed q fever, the only blood laboratory test that affected antibiotic therapy.  

3. "Perhaps it should be separately specified which biochemical analyses did not influence medical care and which bacteriological ones did or did not."

3a. I don't understand that comment. I think we addressed that point.

4. An interesting article and a different perspective. Such observations, if presented, can make a significant difference for the patient in terms of hospital and treatment cost. So, it is nice to address this issue. However, it is my opinion that few more aspects can be studied at length and presented:  

 We thank the reviewer for his assessment and hope there is agreement on including only laboratory blood tests. 

Reviewer 2 Report

Comments and Suggestions for Authors

An interesting article and a different perspective. Such observations, if presented, can make a significant difference for the patient in terms of hospital and treatment cost. So, it is nice to address this issue. However, it is my opinion that few more aspects can be  studied at length and presented:  

1.       Age and sex like parameters are not relevant to study although presented by authors

2.       Others points to look for or evaluate are - 

-          Effect of insurance policy

-          protocol testing (why tests are ordered repeatedly if not important for patient care)

-          policy of hospital

-          clinician point of view or reason for ordering test

-          decision relevant to pk pd of medication, antibiotics administered (some tests may be directly involved in decision making and some indirectly)

-          were the tests done on admission to rule out differential diagnosis ? or were necessary to reach up to the diagnosis

It will be better if data from records related to these points are looked for and analyzed

3.       Authors have written in material and methods that it was a prospective study however, at the end it is written that informed consent was waived off due to retrospective nature of study.

Comments on the Quality of English Language

-

Author Response

  1. Age and sex like parameters are not relevant to study although presented by authors

1a. We believe it is important when comparing studies and for the readers to determine if the results can be extrapolated to their patients. Age and gender are nearly always reported. 

  1. Others points to look for or evaluate are - 

-          Effect of insurance policy

2a. Lines 78-79 we added; All patients are insured and do not pay for hospitalizations.-

  1. protocol testing (why tests are ordered repeatedly if not important for patient care)

-          policy of hospital

3a. Lines 76-77 we added: Except for troponin testing, there is no hospital policy on recommendations for laboratory blood testing -

3b.  clinician point of view or reason for ordering test

Line 262-We added: In our hospital physicians order laboratory tests routinely and not selectively.

  1. decision relevant to pk pd of medication, antibiotics administered (some tests may be directly involved in decision making and some indirectly)

4a. Lines 95-97. The impact of tests on medical management was assessed by chart review and categorized as those that influenced or did not influence decisions about medical care, including intravenous fluids, hypoglycemic drug treatment, or antibiotic therapy.

  1. were the tests done on admission to rule out differential diagnosis ? or were necessary to reach up to the diagnosis –

5a. admission tests were not needed to make the diagnosis. We added;

Line 69-70 The chest x-ray was done because of fever with or without accompanying respiratory tract symptoms.

 Line 176-177  Furthermore, in patients where the admission diagnosis is unclear, other laboratory blood tests can aid in determining the cause of the patient's presentation.

It will be better if data from records related to these points are looked for and analyzed

  1. Authors have written in material and methods that it was a prospective study however, at the end it is written that informed consent was waived off due to retrospective nature of study.

 6a. Line 287; we changed the word retrospective to historical. Furthermore, we also added the date of the approval as requested previously (line 286)